# Adsorption and Desorption of Immune-Modulating Substances by Aluminium-Based Adjuvants: An Overlooked Feature of the Immune-Stimulating Mechanisms of Aluminium-Based Adjuvants

**DOI:** 10.3390/ijms252212399

**Published:** 2024-11-19

**Authors:** Ravi Danielsson, Irene Mile, Håkan Eriksson

**Affiliations:** Department of Biomedical Science, Faculty of Health and Society, Malmö University, SE-205 06 Malmö, Sweden

**Keywords:** aluminium adjuvant, cytokines, inflammation, interleukin, protein corona

## Abstract

Vaccine antigens are partly adsorbed onto aluminium-based adjuvant particles, forming an unstable corona. At the inoculation site, the corona will be restructured, and the adsorbed antigens will be released through replacement with biomolecules from the interstitial fluid of the recipient. Aluminium-based adjuvants (ABAs) carrying a corona of serum proteins as a model of particles with a pre-formed antigen corona were shown to adsorb several categories of cytokines and growth factors, as assessed from a protein array covering 18 different analytes. Out of the 18 analytes, 12 were shown to be adsorbed by the aluminium-based adjuvant Alhydrogel^®^, which had a pre-formed protein corona. The adsorption of TNF-α, IL-2, IL-4, IL-10, and IFN-γ was studied in detail. Among the studied cytokines, IL-2, IL-4, and IFN-γ, were adsorbed by Alhydrogel^®^. Adsorbed IFN-γ was further studied to show that the adsorption of IFN-γ did not denature the cytokine, and the cytokine could be desorbed from adjuvant particles in a biologically active form and in relevant amounts. The adsorption of immune-stimulating molecules onto ABAs at the administration site of a vaccine is a neglected event in the mode of action of aluminium-based adjuvants. This process may modulate the immune response with a profound impact on initiating the innate immune response and consequently the adaptive immune response.

## 1. Introduction

More than 90 years ago, Glenny et al. [1] introduced aluminous compounds as adjuvants in a vaccine preparation, but despite the extensive use of aluminium-based adjuvants (ABAs) ever since, the mechanisms behind the immune-stimulating properties of ABAs have not been fully appreciated.

Several reviews collating the current overall understanding of the immune-stimulating properties of ABAs are available [2,3,4,5,6], describing the recognised inflammatory properties of ABAs and the infiltration of immune-competent cells at the site of vaccine administration. A range of interactions mediates antigen adsorption on the ABA surface, and the mode of action is quite well described in the literature [7,8,9]. The robustness of the antigen–ABA complex is regarded as a crucial factor in stimulating an immune response [10]. For example, strongly adsorbed antigens may trigger a weaker immune response when compared to antigens with an attenuated affinity for ABAs [11,12].

The administration of a vaccine will affect the corona of the ABAs as well as of all other forms of particulate adjuvants. Endogenous proteins and other biomolecules will modify the corona of the ABAs and facilitate the displacement of adsorbed antigens. Recently, the adsorption of calreticulin and HMGB1—known as damage-associated molecular patterns (DAMPs)—by ABAs containing a pre-formed protein corona was reported [13], indicating a continuous rearrangement of the protein corona of the adjuvant particles. Modification of the ABA protein corona at the administration site is generally described as the release of adsorbed antigens rather than competitive adsorption of immune-modulating molecules by the ABAs at the administration site. Thus, the competitive adsorption of biomolecules may be an overlooked segment of the actual chain of events, and, in this paper, the adsorption of immune-polarising cytokines by ABAs possessing a pre-formed corona is communicated, implying a pivotal role of the adsorption of biomolecules by ABAs at the administration site. The quantity of ABAs in a vaccine usually exceeds that of the antigen to ensure quantitative adsorption and hence an adequate supply of antigen. At the inoculation site, the ABA corona is continuously remodelled, resulting in antigen release and the adsorption of biomolecules, e.g., cytokines, induced and produced by local cells. The adsorption and, in essence, the quenching of immune-stimulating molecules at the administration site is a neglected event and may shed light on “the immunologist’s dirty little secret” as stated by Janeway [14], since very few inflammatory and immune regulatory molecules induced by ABAs are reported. The initiation of an innate immune response at the administration site is essentially the cause of an ensured adaptive immune response and thereby reflects the potency of the adjuvant. A better understanding of the mode of action of adjuvants such as ABAs is fundamental in our attempts to create better and safer vaccines.

## 2. Results

### 2.1. Formation of a Protein Corona on ABAs and Identification of Calreticulin Adsorbed by Alhydrogel^®^ Possessing a Pre-Formed Protein Corona

ABAs have a high protein adsorption capacity, and after incubation with serum, a substantial protein layer will be adsorbed onto the surface of the adjuvant particles. The detachment of corona-forming proteins from the ABAs, achieved overnight in a culture medium containing 10% foetal calf serum, can be facilitated by the detergent SDS (Figure 1a). However, no attempts have been made to verify if the total desorption of adsorbed proteins would be obtained by the SDS treatment.

Damage-associated molecular patterns (DAMPs) or alarmins such as calreticulin and HMGB1 have recently been reported to be released by peripheral monocytes co-cultured with ABAs; however, the extracellular presence of the released DAMPs was attenuated due to adsorption by adjuvant particles in the cellular environment [13]. Calreticulin can be detected among the proteins adsorbed by Alhydrogel^®^ upon co-culture with the monocytic cell line THP-1, whereas no adsorption could be observed from Adju-Phos^®^ (Figure 1b). Calreticulin adsorption indicates that although a protein corona is formed upon incubation with serum proteins, it does not completely block the further adsorption of immune-modulating factors.

### 2.2. Cytokines and Growth Factors Adsorbed by ABAs Possessing a Pre-Formed Corona

Cytokines and growth factors are major immune-modulating factors. To screen if these molecules could be adsorbed by ABAs possessing a pre-formed protein corona, a medium from stimulated PBMCs and the Proteome Profiler Human Cytokine Array Kit were used. In the culture medium from the stimulated PBMCs, the Proteome Profiler Array detected 18 different cytokines and growth factors with a pixel density higher than 5% of the positive reference samples (Figure 2). Table 1 shows examples of the spot densities and the ratios of cytokines with high, medium, and low spot densities. The cytokines shown in Table 1 are the three cytokines marked in Figure 2, showing the correlation between appearance on the array membrane and the measured pixel density.

After incubation with two concentrations of Alhydrogel^®^, the ratio of the incubation with 200 μg/mL and 20 μg/mL of Alhydrogel^®^ was calculated. A ratio less than 1 indicated dose-dependent adsorption of the analytes by the ABA, and dose-dependent adsorption by 12 out of the 18 detectable analytes was observed using Alhydrogel^®^ (Table 2). This experiment was repeated three times.

A summary of cytokines and growth factors detected by the cytokine array and their ratio after incubation with two concentrations of Alhydrogel^®^ is shown in Table 2. Analytes with a ratio, including their standard deviation, of less than 1 were regarded as adsorbed by Alhydrogel^®^.

To verify the adsorption of cytokines by ABAs possessing a pre-formed protein corona as shown by the Proteome Profiler Array, IL-2, IL-4, IL-10, IFN-γ, and TNF-α were chosen for further investigation as examples of inflammatory, activating, and polarising cytokines.

Various amounts of ABAs with a pre-formed protein corona were incubated with a culture medium containing added recombinant human cytokines. After incubation, the ABA particles were removed by centrifugation, and the cytokine concentration in the supernatant was determined using ELISA (Figure 3).

Four out of the five cytokines—IL-2, IL-4, IL-10, and IFN-γ—were evidently adsorbed by Alhydrogel^®^, whereas almost no adsorption could be detected by Adju-Phos^®^ (Figure 3). Only slight adsorption of IFN-γ by Adju-Phos^®^ could be observed, while Alhydrogel^®^, on the other hand, almost completely removed IFN-γ from the solution, already at a concentration of 50 μg Alhydrogel^®^/mL (Figure 3D), whereas TNF-α showed no adsorption at all by either Alhydrogel^®^ or Adju-Phos^®^ (Figure 3C). The lack of adsorption of TNF-α by Alhydrogel^®^ was verified by incubating the same cytokine-containing medium that was used in the Proteome Profiler Array with Alhydrogel^®^ and analysing the result with the TNF-α ELISA.

The Proteome Profiler Array showed similar adsorption of IL-2, IFN-γ, and TNF-α (Table 2), whereas the adsorption profiles using recombinant cytokines (Figure 3) were completely different between the three cytokines. Recombinant TNF-α showed no adsorption and IFN-γ showed solid adsorption with the complete removal of IFN-γ from the culture medium. Recombinant proteins produced in *E. coli* pose no post-ribosomal modifications such as glycosylation. Most cytokines, such as IFN-γ, are glycosylated, and glycosylated recombinant IFN-γ showed a completely different adsorption profile, which correlated well to the adsorption profile of IFN-γ obtained from conditioned culture medium containing IFN-γ (Figure 4).

### 2.3. Saturation of Non-Glycosylated IFN-γ by Alhydrogel^®^ and Its Desorption from Alhydrogel^®^

The adsorption of both antigen and IL-12 onto aluminium oxyhydroxide (AlO(OH)) before the immunisation of mice has been reported to induce a polarised type 1 cytokine profile, and the authors suggest the polarisation be due to a prolonged biological half-life of IL-12 at the administration site [15,16]. Due to the high adsorption capacity of Alhydrogel^®^, non-glycosylated IFN-γ was chosen as a model to study whether the adsorption and subsequent release of cytokines could have any biological effects.

No saturation of adsorption at the ng/mL level of non-glycosylated IFN-γ was apparent using Alhydrogel^®^ (Figure 5A). Initially, Alhydrogel^®^ carrying adsorbed non-glycosylated IFN-γ was quickly washed with PBS to remove IFN-γ entrapped in the adjuvant pellet. The adjuvant/IFN-γ complex was then resuspended in culture medium, and the concentration of desorbed IFN-γ was measured. After 15 min, the concentration of IFN-γ was 300–400 pg/mL and remained the same after 24 h.

The desorption profile was explored via repeated resuspensions of Alhydrogel^®^ after the adsorption of IFN-γ (Figure 5B). In the loading process of the Alhydrogel^®^, using a cytokine concentration of 50 ng/mL, more than 95% of the IFN-γ was adsorbed. Before the repeated resuspensions, the adjuvant/cytokine complex was quickly washed with PBS, and the cytokine content of the PBS wash of Alhydrogel^®^ was less than 0.05% of the adsorbed cytokine content. Resuspensions of Alhydrogel^®^ with adsorbed IFN-γ resulted in an almost-constant release of IFN-γ, which increased slightly as the resuspensions proceeded. The slowly increasing release of IFN-γ on consecutive resuspensions was reproducible during repetitions of the experiment. All the resuspensions, including the PBS wash, released only 6% of the adsorbed IFN-γ after the PBS wash and five resuspensions (Figure 5B).

### 2.4. Biological Effects of IFN-γ Adsorbed and Released by Alhydrogel^®^

The adsorption of proteins onto ABAs usually results in a conformational change in the protein [4,17], and it is possible that release from the adsorbent would lead to a denatured protein without any or severely attenuated biological activity. The release of cytokines was analysed using ELISA, but an immune assay does not provide any activity information per se. IFN-γ is known to induce and quickly upregulate the expression of the major histocompatibility complex (MHC) on a variety of cells [18], and the upregulation of MHC was used in the assay of the biological activity of IFN-γ released from ABAs. The cell line THP-1 upregulated the expression of MHC class I and class II upon culture in the presence of IFN-γ (Figure 6 and Appendix A). However, the cells are capable of endocytosis, and to avoid endocytosis of the ABA particles, tissue culture inserts (TC inserts) were used to separate cells from the ABA.

Alhydrogel^®^ with adsorbed IFN-γ was resuspended in R10 culture medium and added to culture wells, and TC inserts containing THP-1 cells were placed in the wells. The TC inserts had a pore size of 5 μm, allowing an unhindered equilibrium of soluble molecules between the medium in the culture wells and the inserts, whereas contact between cells and Alhydrogel^®^ was suppressed. Thus, when cultured in the presence of Alhydrogel^®^ carrying adsorbed IFN-γ, THP-1 cells in the TC inserts upregulated the expression of MHC class I and class II. The induced expression by MHC-II is shown in Figure 6, and that by MHC-1 is shown in Appendix A.

A dose-dependent response of IFN-γ added to the medium was obtained at concentrations ranging from 0.2 ng to 5 ng IFN-γ/mL with a detection limit of 0.1 ng IFN-γ/mL. To study any potential biological activity of IFN-γ adsorbed by Alhydrogel^®^, different concentrations of Alhydrogel^®^ carrying adsorbed IFN-γ were resuspended in a medium and cultured with THP-1 cells in TC inserts. Assuming total recovery of the ABA during the loading process and a total release of all adsorbed IFN-γ, the cytokine content of the different ABA resuspensions would correspond to 11.7 and 2.5 ng IFN-γ, respectively. The THP-1 cells were cultured for 24 h and then removed from the TC inserts. Then, new TC inserts with new portions of THP-1 cells were added to the previously used medium containing added IFN-γ or Alhydrogel^®^; the cells were cultured for an additional 24 h before harvest; and staining was performed with antibodies against MHC class I and class II. No effect on the expression of MHC was noticed on the cultured THP-1 cells in the presence of a high concentration of Alhydrogel^®^—i.e., the addition of 200 μg without adsorbed IFN-γ into 4 mL of culture medium—whereas cells cultured in the presence of Alhydrogel^®^ carrying adsorbed IFN-γ showed a distinct increase in the expression of MHC class I and class II. Upon the addition of 32 μg Alhydrogel^®^ containing adsorbed IFN-γ to the medium, IFN-γ was released, and the concentration remained constant over time, whereas upon the addition of 6.7 μg Alhydrogel^®^, the concentration of IFN-γ released into the medium decreased over time, indicating that the Alhydrogel^®^ became completely depleted of adsorbed IFN-γ (Appendix A). Although the concentration of IFN-γ mediated by release from Alhydrogel^®^ was much lower than the concentration reached upon the addition of recombinant IFN-γ directly to the medium, a corresponding upregulation of the MHC expression was realised. Table 3 shows a summary of the effect on the MHC expression by THP-1 cells after culture in the presence of various concentrations of IFN-γ, Alhydrogel^®^, and Alhydrogel^®^ carrying adsorbed IFN-γ.

## 3. Discussion

The intricacies of how ABAs mediate their immune-stimulating properties are not fully understood. However, the modus operandi is undoubtedly facilitated by several synergistic events, including increased antigen uptake by phagocytosing cells owing to the adsorption of an antigen on ABA particles, and thereby targeting antigen-presenting cells, and the induction of local inflammation and at the same time retaining the antigen at the administration site [3,19]. ABAs are believed to induce an inflammatory response [20,21,22] and are considered as inducers of a Th2 response; consequently, ABAs are weak inducers of Th1 and Th17 responses [4,23,24].

Several genes, including genes coding for inflammatory and polarising cytokines, are induced at the administration site by ABAs in the form of aluminium oxyhydroxide [25] as well as upon in vitro exposure of monocytes/macrophages to aluminium oxyhydroxide [26]. However, at the protein level, very few inflammatory and polarising cytokines are shown to be induced by ABAs, and in principle, only IL-1β and cytokines of the IL-18 and IL-33 families have been reported [27,28]. The release of another group of inflammatory mediators, DAMPs, is not detected in an array of mRNA after exposure to ABAs, since they are already translated and present inside the cells. DAMPs are released upon necrosis and, as reported, are induced by ABAs at the administration site of a vaccine, and immune activation is thereby stimulated [2,29,30]. In this regard, the report that viable peripheral monocytes exposed to ABAs in vitro secrete the DAMP molecules calreticulin and HMGB1, followed by adsorption by the ABAs [13], further emphasises the potential role of the induced secretion and sequential adsorption of immune-modulating factors by ABAs in relation to their immune-stimulating properties. This study shows that an existing protein corona of the ABA does not prevent further adsorption of DAMP molecules (Figure 1); however, DAMP molecules are not the only immune-stimulatory molecules that can be adsorbed onto ABAs possessing a pre-formed corona.

A culture medium from stimulated PBMCs and an array were used to screen potential cytokines and growth factors adsorbed by Alhydrogel^®^ possessing an existing protein corona. The Proteome Profiler Array detected 18 analytes present in the culture medium, and although some of the analytes showed a relatively high standard deviation, a dose-dependent reduction in 12 of the analytes after incubation with two different concentrations of Alhydrogel^®^ was evident (Table 2). The Proteome Profiler Array clearly showed that IL-1β was not adsorbed by Alhydrogel^®^. This result is consistent with previously published results, since IL-1β has been shown to remain in solution upon incubation with Alhydrogel^®^ [13]; i.e., upon cellular secretion, IL-1β is not adsorbed by extracellular Alhydrogel^®^, and IL-1β is one of the few cytokines shown to be induced at the protein level by ABAs in vitro [27,31,32].

To verify the results obtained by the Proteome Profiler Array, adsorption by ABAs of recombinant IL-2, TNF-α, and IFN-γ in culture medium was investigated using ELISA. However, the results from the recombinant cytokine assays did not coincide with the array results. Whereas the adsorption of PBMC-derived TNF-α was indicated by the array, recombinant TNF-α was not at all absorbed by either Alhydrogel^®^ or Adju-Phos^®^ using an ELISA as read-out (Figure 3C). PBMC-derived IL-2 and IFN-γ had similar adsorption characteristics, as indicated by the array, whereas incubations with added recombinant cytokines disclosed completely different adsorption profiles, with almost-quantitative adsorption of IFN-γ by Alhydrogel^®^ (Table 2 and Figure 3A,D).

Recombinant proteins produced by *E. coli* do not pose post-ribosomal modifications such as glycosylation, and glycosylated IFN-γ showed the same adsorption profile as IFN-γ produced by PBMCs (Figure 4).

In a multiplex assay, the assay must differentiate several analytes from the matrix components without analyte interference resulting in false-positive or false-negative outcomes. Although multiplex assays such as the Proteome Profiler Arrays are useful methods in the initial screening process, the results should always be confirmed as in this example using an independent ELISA. The results obtained regarding TFN-α by the Proteome Profiler Array highlights the need for a second opinion in the form of an ELISA, whereas the IFN-γ results highlight the importance of using antigens with the same 3D structure in both the control ELISA and the array.

Although neither IL-4 nor IL-10 was detected in the medium from the stimulated PBMCs, they were included in the investigation as recombinant cytokines (Figure 3B,E). IL-4 showed an adsorption profile like IL-2, whereas IL-10 showed only a slight adsorption by Alhydrogel^®^ at the highest investigated concentrations (Figure 3E). IL-10 has previously been reported to be induced in vivo upon the administration of ABAs [33,34], and ABAs are generally regarded to induce a Th2 response [23].

The adsorption of immune-modulating substances by ABAs, as shown in this paper and when performing in vitro experiments, will make the substances difficult to detect in the medium at the protein level, e.g., using ELISA. This will mask the induction and release of cytokines from cells in response to exposure to ABAs. To our knowledge, the induction of polarising cytokines such as IFN-γ secreted by monocytes/macrophages upon exposure to ABAs in vitro has only been shown at the mRNA level and indirectly by detecting induced proteins using mass spectrometry. Their presence as translated proteins was stated, since an increase in proteins downstream of the cytokines was shown to be augmented using proteomics-enabled mass spectrometry [26]. These techniques are far more sensitive than ELISA, which is the more commonly used technique to identify and quantify cytokines secreted by cells. The low concentrations of cytokines in the culture medium—despite the cytokines being induced at the mRNA level through exposure to ABAs, as reported by Kooijman et al. [26]—indicates that the cytokines induced are sequentially adsorbed by the ABA.

Cytokines adsorbed onto aluminium oxyhydroxide have been reported to have a polarising effect on the in vivo immune response. The adsorption of IL-12 together with an antigen onto aluminium hydroxide before its administration to mice polarised the immune response to a Th1 response [15]. Furthermore, a Th1 immune response has been reported when nanoparticle-sized Alhydrogel^®^ was used as an adjuvant in mice [35]. At neutral pH, the Alhydrogel^®^ forms microparticles, and the Alhydrogel^®^ nanoparticles were obtained by coating the ABA with 2000 Da polyacrylic acid. Upon immunisation, an increased innate expression of IL-12p70, IFN-γ, and IL-1β at the protein level was noted, resulting in a Th1 immune response. Coating of the adjuvant particles resulted in a change in the surface charge of the particles at pH 7, from positive to negative, and this certainly affects the corona of the particle, a change that will lead to a completely different adsorption/desorption behaviour between the micro- and nano-sized Alhydrogel^®^.

If the adsorption/desorption of induced cytokines on extracellular ABAs occurs in vivo, it can be hypothesised that extracellular ABAs will influence the local inflammatory and immune-stimulating response. A burst of cytokines from infiltrating myeloid cells at the administration site of a vaccine will be transformed into a depot of cytokines adsorbed by ABAs, and a peak of cytokines produced by the infiltrating cells will be transformed into an attenuated/modulated but still biologically relevant concentration of cytokines over a prolonged time, sharpening the immune activation.

In this report, the protein corona of ABAs was formed by exposing ABAs to a protein concentration of approximately 10 mg/mL at neutral pH, vide supra. This is well in agreement with the in vivo composition of the interstitial fluid, i.e., 20 mg protein/mL, which the ABAs will be exposed to upon subcutaneous or intramuscular vaccine administration [36].

Alhydrogel^®^ displayed no saturation regarding non-glycosylated IFN-γ at concentrations far exceeding the expected in vivo concentrations (Figure 5A), and the adsorbed IFN-γ could be desorbed and released from Alhydrogel^®^ (Figure 5B). Non-glycosylated IFN-γ was then chosen as a model to study if cytokines adsorbed onto ABAs could mediate an immune-modulating response. The model using non-glycosylated IFN-γ showed an almost constant release of IFN-γ achieved upon repeated resuspensions of IFN-γ-loaded Alhydrogel^®^ (Figure 5B). This indicates that all Alhydrogel^®^ IFN-γ adsorption sites are identical and that IFN-γ desorption is equilibrium driven. Thus, Alhydrogel^®^ will act as the depot of IFN-γ, and the cytokine will be desorbed and released only when it is consumed in the surrounding medium; this was confirmed by the upregulation of MHC by THP-1 cells after co-culture with Alhydrogel^®^ containing adsorbed non-glycosylated IFN-γ (Figure 6, Table 3). These results indicate that the corona of adsorbed biomolecules on ABAs is essential and can be a part of the immune-modulating properties of ABAs.

Essentially, the adaptive immune response is guided and directed by the initial innate immune response, and no common understanding regarding the impact on the adaptive response using ABAs has been reached. The initial native immune response will be influenced by features such as the administration route of the vaccine, the antigen and ABA dosages, and the genetic background of the recipient. Conflicting results regarding the in vivo polarising of the immune response by ABAs have been reported [3,24], and different ways of administering the ABA have been implicated as causing the conflicting results. Different administration routes—e.g., intraperitoneally, intramuscularly, or subcutaneously—and the concentration of ABAs will undoubtedly affect the local accumulation of extracellular ABAs and thereby the potential to affect the extracellular availability of the different polarising cytokines at the inoculation site.

All things considered, the adsorption and desorption of cytokines and other immune-stimulating substances by ABAs may have been overlooked. The dosage of ABAs and the administration route will affect the contribution of local adsorption/desorption events of immune-stimulatory substances, e.g., cytokines, on the induced native immune response at the administration site of a vaccine, and in lymph nodes. The possibilities of adsorption of immune-modulating cytokines by extracellular, not-phagocytosed ABA particles in vivo—and thus, in some aspects, a controlled or delayed release of selected cytokines—add further to the complexity of the immune-stimulatory properties of ABAs.

This study also includes the adsorption of the cytokines IL-2, IL-4, TNF-α, and IFN-γ by Al(OH)x(PO_4_)y in the form of Adju-Phos^®^. Adju-Phos^®^ did not significantly adsorb any of the investigated cytokines (Figure 3), clearly different from Alhydrogel^®^. Aluminium hydroxy phosphate is also regarded as a Th2 inducer; however, compared to aluminium oxyhydroxide, the surface charge is completely different. At neutral pH, Alhydrogel^®^ and Adju-Phos^®^ have opposite charges, and the adsorption of antigen and its use in vaccine formulations is highly regulated by the isoelectric point of the antigen [10]. It is not surprising that the two forms of ABAs interact differently with different proteins, as noted by the different proteins eluted from Alhydrogel^®^ and Adju-Phos^®^ after incubation with a serum-containing culture medium (Figure 1a). This study has mainly addressed Alhydrogel^®^, and so far, no screening has been performed revealing the adsorption of potential immune modulators by Adju-Phos^®^ to identify immune modulators that might polarise the immune stimulation by aluminium hydroxy phosphate.

## 4. Materials and Methods

### 4.1. Materials

Aluminium adjuvant preparations: Alhydrogel^®^ (AlO(OH)) and Adju-Phos^®^ (Al(OH)x(PO_4_)y) were purchased from Brenntag Biosector (Frederikssund, Denmark). Human recombinant cytokines IL-2, IL-4, IL-10, TNF-α, and IFN-γ of research grade were acquired from Miltenyi Biotec (Bergisch Gladbach, Germany), whereas recombinant and glycosylated IFN-γ was obtained from Abcam (Cambridge, UK).

### 4.2. Cell Culture

THP-1 (ATCC TIB-202), obtained from LGC Standards, Teddington, UK, was cultured in RPMI 1640 medium supplemented with 10% heat-inactivated foetal calf serum of EU grade (Gibco, ThermoFisher Scientific, Waltham, MA, USA) and 50 μg/mL of gentamicin (Corning Media Tech, ThermoFisher Scientific). This medium is referred to as R10. Cells were cultured at 37 °C in a humidified atmosphere with 5% CO_2_.

### 4.3. Pre-Coating of ABAs and the Formed Protein Corona

ABAs, at 400 μg/mL, were incubated overnight in R10 medium at 37 °C using 12-well culture plates and 3 mL per well. The conditioned ABAs were resuspended, and 1 mL was withdrawn and collected via centrifugation for 5 min at 13,000× *g*. The ABAs were quickly washed via resuspension in 1 mL PBS and collected via centrifugation for 5 min at 13,000× *g*. Finally, the ABAs were resuspended in 100 μL of Laemmli protein sample buffer containing DTT (Bio-Rad Laboratories, Hercules, CA, USA); proteins forming the ABA’s corona were identified by SDS-PAGE using NuPAGE 4–12% Bis-Tris gel (1.0 mm × 15 wells, Invitrogen, ThermoFisher Scientific and SDS-Running Buffer (NuPAGE MES SDS Running Buffer, Invitrogen, ThermoFisher Scientific). Samples and molecular-weight markers (see Blue Pre-Stained Protein Standard, Invitrogen, ThermoFisher Scientific) were subjected to a constant voltage of 180 V. Protein bands were visualised using Coomassie Brilliant Blue R-250 staining solution (Bio-Rad Laboratories). Images of the SDS-PAGE were captured using a FluorChem E System (ProteinSimple, San Jose, CA, USA).

### 4.4. Identification of Adsorbed Calreticulin

A total of 3 mL of THP-1 cells, 1 × 10^6^/mL, in R10 were incubated overnight at 37 °C in an equal volume of not-pre-coated ABAs, 200 μg/mL in R10, and harvested via centrifugation, 10 min at 10,000× *g*. The pellet containing ABA and the cells was resuspended in 1.5 mL distilled water for 5 min to disrupt co-sedimented cells, and the samples were centrifuged for 5 min at 20,000× *g*. The collected pellet was again resuspended in distilled water (1 mL) and centrifuged, this time for 10 min at 10,000× *g*, and finally resuspended in 100 μL SDS-PAGE Sample Buffer containing DTT. Complete lysis of the cells after resuspensions in distilled water was confirmed via analysis using flow cytometry. As a negative control, ABAs in R10 culture medium were incubated overnight at 37 °C. As a positive control, 6 mL of THP-1 cells, 0.5 × 10^6^/mL in R10, were incubated overnight at 37 °C. The cells were collected and solubilised in 800 μL SDS-PAGE Sample Buffer containing DTT. Samples, 10 µL, were separated using SDS-PAGE and transferred to 0.45 μm PVDF membranes (Invitrogen, ThermoFisher Scientific) via semi-dry blotting and NuPAGE Transfer Buffer (Invitrogen, ThermoFisher Scientific). Membranes were washed with TBS-Tween, blocked with 5% (*w*/*v*) dried milk powder in TBS-Tween, and incubated overnight at 8 °C with anti-calreticulin (R&D Systems, Minneapolis, MN, USA), 1 μg/mL, in TBS-Tween-milk powder. The membrane was stained for one hour at room temperature with HRP-Goat anti-Mouse IgG (ThermoFisher Scientific), 1 μg/mL, in TBS-Tween-milk powder and visualised using SuperSignal^TM^ West Pico Plus Chemiluminescent Substrate (ThermoFisher Scientific). Images were captured using the FluorChem E System (ProteinSimple).

### 4.5. Preparation of Cytokine-Containing Medium

Peripheral blood mononuclear cells (PBMCs) were obtained via density centrifugation on Ficoll-Paque^TM^ (GE Healthcare Life Sciences, Uppsala, Sweden) of a leukocyte concentrate purchased from Blodcentralen, Skånes universitetssjukhus, SUS, Lund, Sweden, and PBMCs, 2 × 10^6^ cells/mL, were cultured overnight in R10 medium containing eBioscience^TM^ Cell Stimulation Cocktail (Invitrogen, ThermoFisher Scientific) at 37 °C. The medium was collected, clarified via centrifugation, 10 min at 13,000× *g*, and stored at −80 °C until use.

### 4.6. Adsorption of Cytokines by ABAs

#### 4.6.1. PBMC-Cytokine-Containing Medium

Alhydrogel^®^, 400 μg/mL and 40 μg/mL, pre-incubated in R10 to form a protein corona, were incubated overnight at 37 °C with an equal volume of cytokine-containing medium from PBMCs, vide supra. Incubations were performed in a total volume of 200 μL per well in a 96-well plate. As control, the cytokine-containing medium was diluted with R10 and incubated. Six wells of each sample were incubated, pooled, cleared via centrifugation, for 10 min at 13,000× *g*, and stored at −80 °C until assayed via the human cytokine array.

#### 4.6.2. Medium Containing Recombinant Cytokines

ABAs, 400 to 25 μg/mL, pre-incubated in R10 to form a protein corona, were incubated for a minimum of 2 h at 37 °C with an equal volume of R10 medium containing the recombinant human cytokines, IL-2, IL-4, IL-10, IFN-γ, or TNF-α made from stock solutions of 10 μg/mL to obtain the following concentrations: IL-2, 3 ng/mL; IL-4, 4 ng/mL; IL-10, 3 ng/mL; TNF-α, 1 ng/mL; and IFN-γ, 2 ng/mL. Actual cytokine concentrations after dilution were determined using ELISA and corresponded with the cytokine concentration obtained after incubations without any addition of ABAs as shown in the figures and tables. Incubations were performed in 96-well plates in triplicate in a total volume of 200 μL per well; incubations with IFN-γ were performed in Eppendorf tubes to avoid unspecific adsorption of the cytokine, since the concentration of non-glycosylated IFN-γ determined using ELISA did not match the expected concentration after the addition of an equal volume of culture medium in 96-well plates. The triplicates were pooled, cleared via centrifugation, 10 min at 13,000 × g, and stored at −80 °C until assayed using ELISA.

As control, cytokines were incubated in R10 medium without any addition of ABAs. Cytokine amounts adsorbed by ABAs were calculated from the difference between cytokine content in R10 incubations and cytokine content after incubation with ABAs.

### 4.7. Proteome Profiler Human Cytokine Array Kit

After the incubation of supernatants from PBMCs stimulated with eBioscience^TM^ Cell Stimulation Cocktail (cytokine-containing medium) with Alhydrogel^®^ (400 μg/mL and 40 μg/mL) or R10 medium, the ABA was removed via centrifugation. One millilitre of each of the cleared supernatants was assayed using the Proteome Profiler Human Cytokine Array Kit (R&D Systems) according to the manufacturer’s instructions. Images of the membranes were captured using the FluorChem E System, and spot densities were analysed according to the manufacturer’s instructions using the Wester Vision Software HLImage^++^ version 1.0.0.1. Each array contained positive reference spots, and analytes possessing 5% or higher spot densities relative to the reference spots were considered as positive.

### 4.8. Cytokine ELISA

The human cytokines IL-2, IL-4, IL-10, IFN-γ, and TNF-α were analysed using DuoSet ELISA (R&D systems), performed according to the manufacturer’s instructions.

### 4.9. Release of Cytokines Adsorbed by ABA

Alhydrogel^®^, 400 μg/mL, pre-incubated in R10, was incubated for 2 h at 37 °C with an equal volume of 100 ng IFN-γ/mL in R10 medium. Incubations with IFN-γ were performed in Eppendorf tubes, 200 μL per tube. The triplicates were pooled and centrifuged for 10 min at 13,000× *g*. The supernatants were stored at −80 °C until the ELISA assay.

The adjuvant pellets were quickly washed via resuspension in 1 mL of PBS and collected via centrifugation for 5 min at 13,000× *g*. The washed adjuvant pellets were resuspended in 600 μL R10 medium, and after 15 min at 37 °C, the samples were centrifuged for 5 min at 13,000× *g*. Resuspension in R10 was repeated until 5 resuspensions had been performed. All supernatants were collected and stored at −80 °C until the ELISA assay.

Cytokines incubated in R10 medium and Alhydrogel^®^ incubated with R10 were employed as controls. The Alhydrogel^®^ control was washed and resuspended as previously described to make sure that the Alhydrogel^®^ had no impact on the ELISA.

### 4.10. Bioassay of IFN-γ

Alhydrogel^®^, 400 μg/mL, pre-incubated in R10, was incubated for 2 h at 37 °C with an equal volume of 100 or 150 ng/mL non-glycosylated IFN-γ in R10. Triplicates were made in Eppendorf tubes in a total volume of 200 μL per tube. The triplicates were pooled and centrifuged for 10 min at 13,000× *g*, and the supernatants were stored at −80 °C until the ELISA assay. The adjuvant pellet was quickly washed via resuspension in 1 mL of PBS, collected via centrifugation for 5 min at 13,000× *g*, and resuspended in R10 medium. In a 6-well plate, 4 mL/well of R10 containing various amounts of resuspended Alhydrogel^®^ were added to the plate (the amounts of Alhydrogel^®^ were based upon a 100% recovery of Alhydrogel^®^ after loading and washing). Pre-wet tissue culture (TC) inserts, pore size 5 μm (Sarstedt, Nümbrecht, Germany), were added to the wells, and 1 mL of THP-1 cells, 1 × 10^6^/mL in R10, were loaded into the TC inserts. After overnight incubation at 37 °C in a humidified atmosphere with 5% CO_2_, the TC inserts were removed from the culture plate, and the cells were collected. The collected cells were resuspended in 1 mL PBS containing 0.1% (*w*/*v*) BSA and 0.1% (*w*/*v*) human IgG, sub-divided into aliquots, 100 μL, and stained with APC-labelled anti-human MHC class I and FITC-labelled anti-human MHC class II (FITC anti-human HLA-DR, DP, DQ, BD Bioscience, San Jose, CA, USA) or APC- and FITC-labelled isotype controls (Miltenyi Biotec and BD Bioscience) via incubation for 30 min on ice. After washing (PBS 0.1% BSA), the cells were resuspended in 250 μL 1% (*w*/*v*) paraformaldehyde and analysed using flow cytometry (Accuri C6 flow cytometer and standard settings).

New TC inserts were added to the previously used culture plate (containing the previously used medium), and the TC inserts were supplemented with a new 1 mL portion of THP-1 cells, incubated overnight at 37 °C, before the cells were collected and stained with antibodies against human MHC class I and MHC class II.

As controls, THP-1 in TC inserts were incubated in wells containing only R10 medium and in wells containing Alhydrogel^®^ without any adsorbed IFN-γ. As positive control, THP-1 (1 mL, 1 × 10^6^/mL) in TC inserts was incubated in wells with 4 mL/well of R10 medium containing recombinant non-glycosylated IFN-γ.

## 5. Conclusions

Upon administration of a vaccine, endogenous proteins of the interstitial fluid, and other biomolecules, will probably bind to and modulate the corona of ABAs. It will likely also show different corona-binding proteins depending on the ABAs chosen for the vaccine, as calreticulin showed in this study for Adju-Phos^®^ and Alhydrogel^®^. This study shows also that an (pre-)established protein corona on Alhydrogel^®^ does not further prevent the adsorption of cytokines and growth factors in vitro. In an in vitro model study, several cytokines were adsorbed by Alhydrogel^®^ possessing a pre-formed protein corona. Adsorbed protein was desorbed from Alhydrogel^®^ by an equilibrium shift as shown by the adsorption/desorption of IFN-γ. IFN-γ was not denatured upon adsorption by Alhydrogel^®^ and was desorbed from the adjuvant particles in a biologically active form, indicating a general adsorption and desorption of cytokines in a biologically active form.

The adsorption of immune-stimulating molecules at the administration site of a vaccine can be hypothesised as an element in the mode of action by ABAs. This is a neglected mode of action by ABAs, which can be assumed to modulate the immune response with a profound impact on initiating the innate immune response and thereby the adaptive immune response.

## Figures and Tables

**Figure 1 ijms-25-12399-f001:**
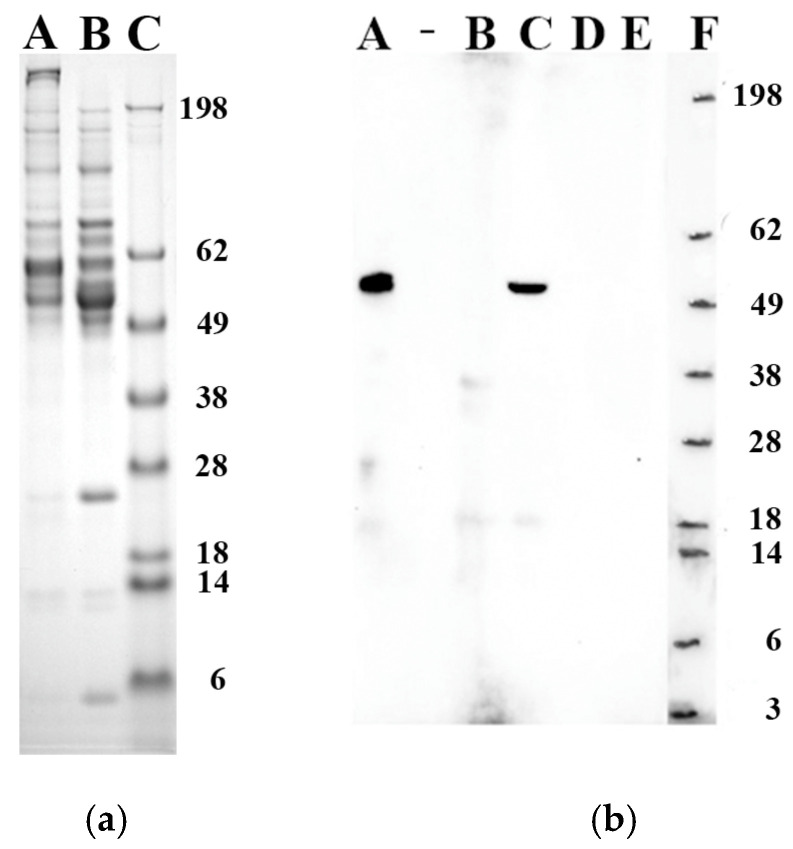
Formation of a protein corona on aluminium based adjuvants (ABAs) upon incubation with foetal calf serum and adsorption of calreticulin. (**a**) Formation of a protein corona; ABAs at a concentration of 400 μg/mL were incubated overnight in a cell culture medium containing 10% FCS (R10). The ABAs were collected, and the proteins adsorbed by the ABAs were analysed using SDS-PAGE. (**A**) Adju-Phos^®^ corresponding to 40 μg. (**B**) Alhydrogel^®^ corresponding to 5 μg. (**C**) Molecular-weight markers with sizes in kDa. (**b**) Western blot showing calreticulin adsorbed by ABAs. THP-1 cells were co-cultured with ABAs, 100 μg/mL, and after overnight co-culture, cells and ABA particles were collected via centrifugation. The co-sedimented cells were lysed, and the ABA particles were washed before the proteins adsorbed by the ABA particles were eluted and separated on SDS-PAGE, and the presence of calreticulin was detected using Western blot and an antibody against calreticulin. (**A**) Lysate of THP-1 cells. (**B**) Adsorption by Adju-Phos^®^ (Brenntag Biosector, Frederikssund, Denmark)corresponding to 30 μg ABA. (**C**) Adsorption by Alhydrogel^®^ corresponding to 30 μg ABA. (**D**) Adju-Phos^®^ medium control. (**E**) Alhydrogel^®^ medium control. (**F**) Molecular-weight markers with sizes in kDa.

**Figure 2 ijms-25-12399-f002:**
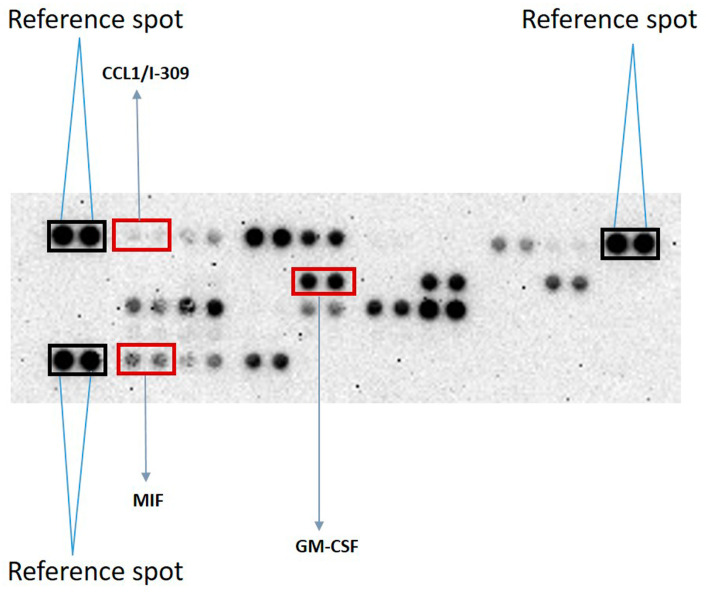
Detection of cytokines/growth factors in the medium from stimulated PBMCs by the human cytokine array. The culture medium from PBMCs stimulated with PMA and ionomycin showed 18 detectable cytokines and growth factors using the Proteome Profiler Human Cytokine Array Kit and HLImage^++^ software (version 1.0.0.). The array shows duplicates of each cytokine and growth factor. Three positive reference spots are shown at the top left, top right, and lower left, marked with rectangles, and examples of cytokines with high, medium, and low spot intensities are marked. Appendix A shows the position (coordinates) at the membrane of all cytokines/growth factors included in the Proteome Profiler Human Cytokine Array Kit.

**Figure 3 ijms-25-12399-f003:**
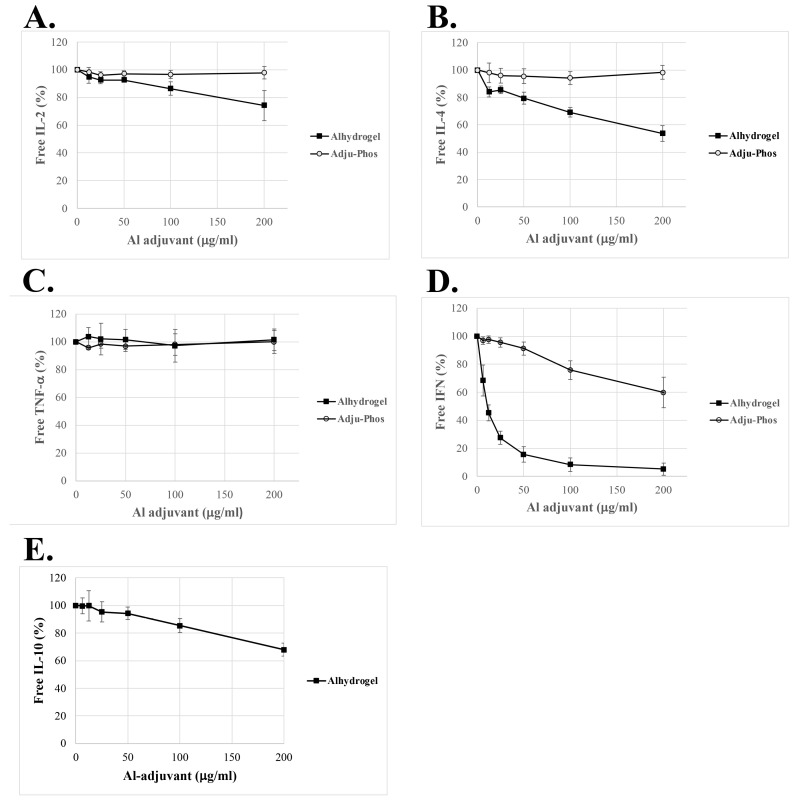
Adsorption of cytokines onto ABAs possessing a pre-formed protein corona. Cell culture media with added recombinant human cytokines (IL-2, IL-4, IL-10, TNF-α, and non-glycosylated IFN-γ) were incubated with various amounts of Alhydrogel^®^ (■) or Adju-Phos^®^ (◯) possessing pre-formed protein corona. After overnight incubation, the adjuvant particles were removed by centrifugation, and the cytokine concentrations in the cleared medium were determined using ELISA. (**A**) IL-2, (**B**) IL-4, (**C**) TNF-α, (**D**) IFN-γ, and (**E**) IL-10. The free cytokine is expressed as % of the cytokine in the control, i.e., incubation without any addition of Alhydrogel^®^. The figure shows the average of three independent experiments and the standard deviation.

**Figure 4 ijms-25-12399-f004:**
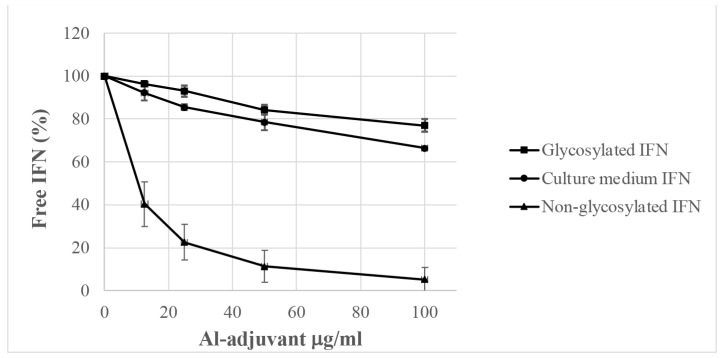
Adsorption of glycosylated and non-glycosylated IFN-γ by Alhydrogel. Alhydrogel^®^ with a pre-formed protein corona of serum proteins was incubated with glycosylated IFN-γ (■), non-glycosylated IFN-γ (▲), and culture medium from PBMCs stimulated with PMA and ionomycin containing IFN-γ produced by the PBMCs (●). After overnight incubation, the adjuvant particles were removed via centrifugation, and the cytokine concentrations in the cleared medium were determined using ELISA. The unbound IFN-γ is expressed as % of the IFN-γ in the control, i.e., incubation without any addition of Alhydrogel^®^. The figure shows the average of three independent experiments and the standard deviation.

**Figure 5 ijms-25-12399-f005:**
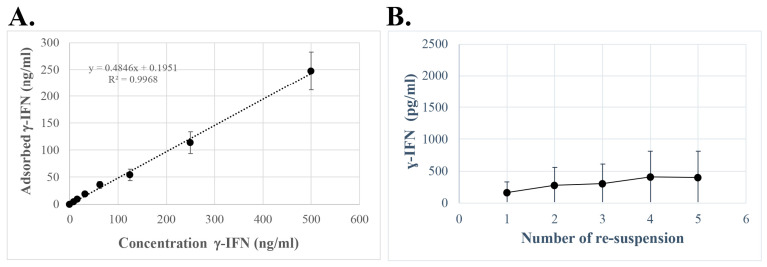
Adsorption and desorption of non-glycosylated IFN-γ by Alhydrogel. (**A**) Adsorption of non-IFN-γ. Alhydrogel^®^ with a pre-formed protein corona of serum proteins was incubated with IFN-γ at a concentration range from 16 ng/mL to 500 ng/mL, and the adsorption of the cytokine was determined using ELISA. Trendline shown as dotted line. (**B**) Release of adsorbed cytokine. Alhydrogel^®^ carrying adsorbed IFN-γ was repeatedly resuspended in R10 culture medium. The supernatants were collected, and the concentration of IFN-γ was determined using ELISA. The figure shows the average of three independent experiments and the standard deviation.

**Figure 6 ijms-25-12399-f006:**
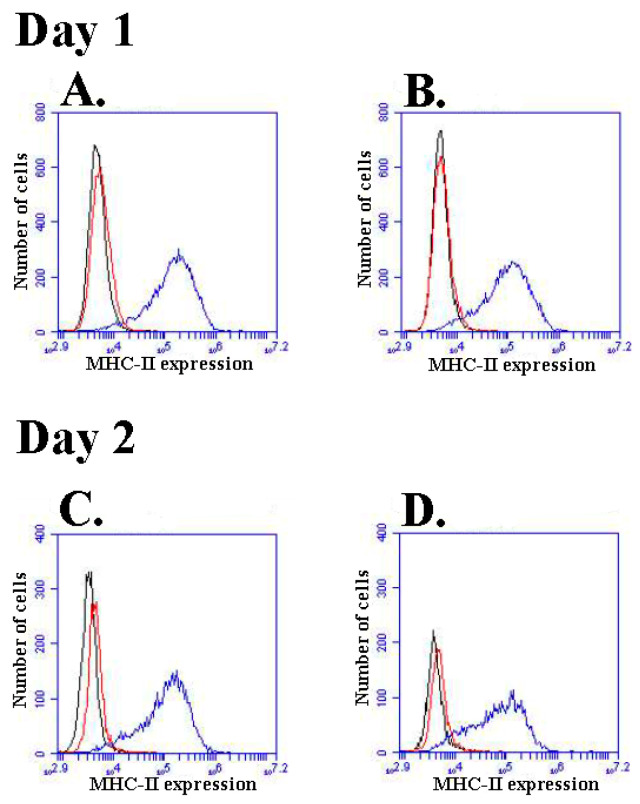
Induced expression of MHC-II by IFN-γ adsorbed onto Alhydrogel^®.^ The THP-1 was cultured overnight, and on day 1, it was removed from the TC inserts and stained with antibodies against MHC class II. A new TC insert with new cells was added to the previously used medium. The cells were cultured overnight, and on the next day (day 2), they were removed from the TC inserts and stained with antibodies against MHC class II. Histograms of THP-1 cells showing the expression of MHC (**A**,**B**) after day 1 and (**C**,**D**) on day 2; A and C: Addition of 37.5 ng IFN-γ into the culture medium, giving a concentration of 8.6 ng IFN-γ/mL, and staining with an isotype control (black histogram) and staining with an FITC-labelled antibody against human MHC class II (blue histogram). Cells cultured without any addition of IFN-γ and stained with an FITC-labelled antibody against human MHC class II (red histogram); B and D: Addition of 32 μg Alhydrogel^®^ containing, at most, 11.7 ng of adsorbed IFN-γ, and staining with an isotype control (black histogram) and staining with an FITC-labelled antibody against human MHC class II (blue histogram). Cells cultured after the addition of 200 μg Alhydrogel^®^ containing no adsorbed IFN-γ and stained with an FITC-labelled antibody against human MHC class II (red histogram). Figure shows representative results from one of three independent experiments performed.

**Table 1 ijms-25-12399-t001:** Examples of analytes with high, medium, and low pixel density after analysis using the Proteome Profiler Human Cytokine Array Kit.

	Experiment 1		Experiment 2		Experiment 3		
Spot	ABA High	ABA Low	ABA High	ABA Low	ABA High	ABA Low	Average High/Low ± SD
GM-CSF	10,903.02	17,328.6	9951.94	12,525.28	9457.14	13,761.82	0.70 ± 0.08
CCL1/I-309	1054.65	624.89	310.91	383.69	3385.27	5344.73	1.04 ± 0.56
MIF	3514.08	7793.85	4129.54	12,224.48	11,790.3	13,751.68	0.55± 0.27

Pixel densities and average ratio of GM-CSF, MIF, and CCL1 obtained from three independent experiments in which a cytokine-containing medium was incubated with a high (200 μg/mL) and a low (20 μg/mL) concentration of Alhydrogel^®^.

**Table 2 ijms-25-12399-t002:** Cytokines and growth factors detected in the medium from stimulated PBMCs by the array and adsorption by Alhydrogel^®^.

Cytokines and Growth Factors Detected in the Culture Medium	Alhydrogel^®^ High/Low Ratio	Adsorbed by Alhydrogel^®^
CCL1/I-309	1.04 +/− 0.56	No
**MCP-1/CCL2**	**0.46 +/− 0.18**	**Yes**
**MIP-1α/MIP-1β, CCL3/CCL4**	**0.84 +/− 0.13**	**Yes**
**RANTES/CCL5**	**0.22 +/− 0.10**	**Yes**
**CXCL1/GRO alpha**	**0.57 +/− 0.18**	**Yes**
CXCL12/SDF-1α	0.58 +/− 0.47	No
**GM-CSF**	**0.70 +/− 0.08**	**Yes**
**IFN-γ**	**0.60 +/− 0.33**	**Yes**
IL-1 beta/IL-F2	1.14 +/− 0.39	No
**IL-1ra/IL-1F3**	**0.56 +/− 0.08**	**Yes**
**IL-2**	**0.52 +/− 0.16**	**Yes**
**IL-5**	**0.39 +/− 0.16**	**Yes**
IL-6	1.19 +/− 0.44	No
IL-8/CXCL8	1.05 +/− 0.08	No
IL-16	0.71 +/− 0.75	No
**MIF**	**0.55 +/− 0.27**	**Yes**
**Serpin E1**	**0.48 +/− 0.30**	**Yes**
**F-α**	**0.47 +/− 0.35**	**Yes**

The medium from stimulated PBMCs was incubated with Alhydrogel^®^ 200 μg/mL or 20 μg/mL, and after removal of the aluminium-based adjuvant (ABA) by centrifugation, the supernatants were analysed by the Proteome Profiler Human Cytokine Array Kit. The spot densities were measured, and the ratio between Alhydrogel^®^ 200 μg/mL (high) and 20 μg/mL (low) was calculated. Bold text shows cytokines regarded to be adsorbed by the Alhydrogel The table shows the average of three independent experiments and the standard deviation.

**Table 3 ijms-25-12399-t003:** THP-1 cells cultured in tissue culture (TC) inserts and stained with antibodies against human MHC class I (anti-HLA-ABC) and against MHC class II (anti-HLA-DR/DP/DQ) after co-culture with various concentrations of directly added IFN-γ or Alhydrogel^®^ containing adsorbed γ-IFN. The expression of MHC is presented as the ratio between mean fluorescence intensity (MFI) after staining with labelled antibody against MHC from cells cultured in the presence of IFN-γ or Alhydrogel^®^ and the MFI obtained after staining THP-1 cells cultured in the TC insert and medium (R10 control). The table shows representative results from one of three independent experiments performed.

Sample	Day 0 Added IFN-γ (ng)	Day 1 Ratio: MFI Sample/MFI R10		Day 2 Ratio: MFI Sample/MFI R10	
		MFI R10	MHC-II	MHC-I	MHC-II
Medium control	0	1.0	1.0	1.0	1.0
Free IFN-γ	7.5	1.3	18.7	1.8	12.7
Free IFN-γ	37.5	1.4	27.8	1.8	24.6
200 μg Alhydrogel^®^	0	1.0	1.0	1.0	1.0
6.7 μg Alhydrogel^®^ with adsorbedIFN-γ	2.5 ^a^	1.3	9.7	1.6	5.0
32 μg Alhydrogel^®^ with adsorbed IFN-γ	11.7 ^a^	1.4	21.3	1.9	18.5

^a^ Calculated concentration assuming 100% recovery of Alhydrogel^®^ after loading with IFN-γ and washing.

## Data Availability

Data available upon request.

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
