# Peer review of "Adsorption and Desorption of Immune-Modulating Substances by Aluminium-Based Adjuvants: An Overlooked Feature of the Immune-Stimulating Mechanisms of Aluminium-Based Adjuvants"

_ijms, 2024, doi:10.3390/ijms252212399_

Round 1
Reviewer 1 Report
Comments and Suggestions for Authors
In the present manuscript Danielsson et al try to identify the mechanisms of aluminium based adjuvants (ABAs). ABAs are known for their adjuvanticity but their mechanisms of action are not fully elucidated. Interestingly despite the fact that are capable of inducing strong innate responses, the levels of cytokines at the site of injection are rather low to zero. Thus, it was hypothesized that ABAs have the capability to adsorbe several biological factors after releasing their antigenic cargo, forming a depot of cytokines and other biological molecules capable of inducing and/or enhancing adaptive immune responses. Based on that hypothesis, authors via different approaches showed that ABAs and specifically Alhydrogel having a pre-formed protein corona were capable of adsorbing several biological factors, i.e. DAMPs or cytokines using in vitro culture systems providing information regarding their mechanism of action. Overall, the theme of the paper is of great interest and the experimentation is well-designed However, the paper's presentation needs further improvement. Specifically,
1. General comment: All Figure legends need tobe written again in a more concise way giving in the introductory sentence the theme of the figure and then explain the results depicted. Additionally, statistic analysis needs to be added in each figure, figure legend as well as in materials and methods section. Were the experiments performed once with multiple technical replicates or the results were extracted from multiple biological replicates, which is the most right thing to do?
2. General comment regarding Results section. Results should be written again in a more descriptive way. Also the figures need to be changed with new ones having greater analysis.
3. In figure 1a, authors should also run R10 medium in order to see if there are differences in the protein motif as well as in the proteins adsorbed, i.e their size, among Alhydrogel and AdjuPhos. In figure 1b, AdhuPhos seems not to adsorb calreticulin. this should be presented in the text of results section (paragraph 2.1), since it seems there is a discrepancy among the two ABAs and they discuss it in the Discussion section.
4. In paragraph 2.2 and also in the respective paragraph of Materials and Methods section, authors shoul specify the target cytokines and growth factors detected by Protein Array. Also, did authors normalized Alhydrogel results with medium only, which is the most proper thing to do? Additionally, in the respective Figure 2, they should match the dots with the cytokines, making the figure more explicable regarding the cytokine/growth factors levels.
5. In lines 180-182, authors should explain the reasons for using the recombinant cytokines panel for conducting ELISA experiments? Was there a biological meaning for selecting those cytokines?
6. In line 199, authors wrote that "..IL-10 showed a low degree of adsorption..". However, the figure did not support it, since the reduction levels seemed quite similar with those of IL-2 and IL-4 as well. Please specify this.
7. In line 241, authors should change the phrase into "Adsorption of non-glycosylated IFNg".
8. Figure 6 shold be chnaged into a more descriptive one. Specifically, authors should present medium control, positive control and then ABAs sample. For reasons of clarity, the histograms shoulb be presented as an overlapping histogram image with different colours, since this it would be better for the readers to discriminate the changes among different cultivation status. Additionally, figure legend should be written in a more concise way. If the lines 284-294 belong to the respecitve legend, they should incorporate in it.
9. Table 3 along with its legend should be changed into a more appropriate one.
10. In paragraph 4.4, in line 516 it is not specified if ABAs used had a pre-formed protein corona. Also, in line 525 authors should specify the exact amount or protein loaded in the gel for each sample. In lines 530 and 532, they should provide the concentration of dry skimmed milk.
11. In line 621, authors wrote that ".. the cells were resuspended in 250 microl 1% w/v paraformaldehyde and analyzed by flow cytometry...". Usually, fixing precedure lasts about 30 min followed by washing to extract any fixative prior analysis. Was this the exact protocol authors followed?
12. Conclusion paragraph should be written again containing all the information taken by the research conducted in the present manuscript and not focused only in the IFNgamma. Additionally, the first two lines present a hypothesis which requires a proof-of-concept (the aim of the present manuscript). Thus, it should be written in such a way, and not as something established. If so, then authors should provide the respective reference.
Author Response
Please find attached a word document with the reviewer's comments and our responce

Reviewer 2 Report
Comments and Suggestions for Authors
This is a well-written work, where materials & methods and results are clearly described and commented. This work adds an important contribution to the interpretation of the mechanism of action of the oldest and still most used vaccine adjuvant. This reviewer congratulates the Authors, who demonstrate great expertise on this topic, and has no further comments, so according to this reviewer the manuscript can be accepted in its current form.
Author Response
We are greatful that the reviewr found the manuscript interesting and that the changes made in the resubitted version have further improved the manuscript.